# Semi-Quantitative Biosecurity Assessment Framework Targeting Prevention of the Introduction and Establishment of *Salmonella* Dublin in Dairy Cattle Herds

**DOI:** 10.3390/ani13162649

**Published:** 2023-08-17

**Authors:** Lars Pedersen, Hans Houe, Erik Rattenborg, Liza Rosenbaum Nielsen

**Affiliations:** 1Department of Veterinary and Animal Sciences, Section for Animal Welfare and Disease Control, University of Copenhagen, 1870 Frederiksberg, Denmark; houe@sund.ku.dk (H.H.); liza@sund.ku.dk (L.R.N.); 2SEGES Innovation P/S, 8200 Aarhus, Denmark; era@seges.dk

**Keywords:** biosecurity, tools, *Salmonella*, prevention, cattle, dairy herds

## Abstract

**Simple Summary:**

The importance of implementing protective measures is related to herd size because of potentially more external contact and the ability of each animal to get in contact with more animals. Therefore, with an increasing number of animals in cattle herds, biosecurity is becoming increasingly essential. In particular, cattle-adapted diseases, such as *Salmonella* Dublin, can survive in the environment for an extended period after it has been excreted in the faeces of infectious animals. Therefore, indirect spread mechanisms, such as the movement of bacteria by contaminated equipment from farm to farm with the risk of contaminating the cow environment and food before oral intake by other animals, become more important. Additionally, the ability to survive in the environment can enhance the risk of the bacteria becoming established as an infection within the herd if improper management procedures compromise hygiene. In this study, we designed an approach and a tool based on existing knowledge, which could be used in further research studies to evaluate the risk of some dairy herds becoming infected compared with others. 'The assessment can be performed through observation of farms (~1 h) and interviews with the farmers (~1 h). This can help reduce the spread of harmful bacteria in dairy cattle production.

**Abstract:**

An increasing average herd size and complexity in farm structures call for a higher level of biosecurity. It can reduce the risk of introducing and establishing pathogens with multiple-pathway and indirect spread mechanisms, such as *Salmonella* Dublin, a pathogen with an increasing occurrence in dairy cattle farms across different countries and continents. Therefore, this study aimed to use existing knowledge to develop a framework with a supporting tool allowing for a time-efficient, yet comprehensive, assessment of biosecurity measures that can help prevent the introduction and establishment of *S.* Dublin in dairy herds. Based on the literature review, a seven-step biosecurity assessment framework was developed and evaluated in collaboration with biosecurity experts. The resulting framework includes a weighted semi-quantitative assessment method with a scoring guide in an electronic supporting tool for 12 biosecurity sections assessed through on-farm observations and farmer interviews. The framework and tool provide a novel approach to comprehensively assess the overall (mainly external) on-farm biosecurity level by a trained biosecurity assessor. They can be used for systematic data collection in epidemiological studies on risk factors for the introduction and establishment of *S.* Dublin in dairy farms. Preliminary interrater reliability testing indicated moderate reliability between assessors with varying biosecurity skills.

## 1. Introduction

In 2016, the European Parliament and the Council adopted the European Union (EU) Animal Health Law [1], laying down requirements for disease prevention and biosecurity in livestock. The need for biosecurity regarding the control of infectious diseases is highlighted by the fact that the average herd size increases in many countries, enhancing the potential risk of introduction and spread of infectious diseases, including zoonotic infections, such as salmonellosis, one of the most common and serious zoonotic diseases in the EU [2,3]. The genus *Salmonella* affects a broad range of animals and represents approximately 2600 serotypes, including *Salmonella enterica* subsp. *enterica* serovar Dublin (*S.* Dublin), which is host-adapted to cattle [4,5] and the most frequently reported serotype in European cattle [6]. *S.* Dublin is receiving increasing attention in several countries due to its severity as a zoonotic hazard [7,8,9], the animal welfare effects [10,11], and economic losses for the cattle sector [12,13,14]. Furthermore, increasing *S.* Dublin occurrence in cattle has been reported in different countries [15,16,17,18]. *S.* Dublin is endemic in the cattle populations of many countries, and a few have initiated serological surveillance and control programmes for *Salmonella* in their cattle sectors [19,20]. More countries have control programmes for clinical salmonellosis in bovines but not active surveillance based on serology [21]. Denmark initiated a surveillance programme targeting *S.* Dublin in 2002. The programme was turned into an eradication programme in 2008 with increasingly stricter control measures implemented over time—e.g., as seen in the changed legislation over time, including the highly restricted movement of cattle out of ‘likely infected herds’ for live purposes [19]. Simultaneously, there is also a strategy to improve internal, external, and national biosecurity in the Danish cattle sector to prevent the introduction and spread of infectious diseases. Despite the national eradication programme for *S.* Dublin and regulations about mandatory actions to address biosecurity in all cattle herds under veterinary herd health consultancy agreements, (>80% of all Danish dairy herds), an increasing prevalence of dairy herds, are classified as ‘likely infected’ in the *S.* Dublin programme, a disturbing trend observed since 2015. Overall, 11.7% of the 2331 dairy herds could not be declared ‘likely free from *S.* Dublin’ and 9.7% of them were antibody-positive in the bulk tank milk testing programme on 19 April 2023 [22]. This, together with an increasing average dairy herd size (217 cows with a production of 10,518 kilo milk per cow in 2021) and the majority of Danish dairy cattle clustered in the northern, western, and southern part of the Jutland peninsula [23,24], suggests that biosecurity needs to be improved to prevent the introduction and further spread (establishment) of *S.* Dublin on dairy farms. However, some farmers are reluctant to implement and control strict biosecurity measures [25]. Some lack motivation [26]. Others do not find biosecurity measures beneficial [27]. It has been suggested that farmers’ motivation may be compromised due to a lack of quantification of biosecurity measures’ effect in and between cattle herds [28]. In addition, *S.* Dublin transmission occurs despite the implemented trade restrictions, and indirect transmission is indicated by the major hazards of cattle herds continuously being or becoming test-positive in endemically infected areas [29,30,31]. This is supported by epidemiological studies conducted in comparable cattle populations in other countries [32,33]. Additionally, whole-genome sequencing analysis has identified the geographical clustering of *S.* Dublin genetic groups [34,35,36,37,38]. Together, this suggests that indirect spread mechanisms are the drivers for the local spread of *S.* Dublin between herds. Therefore, there appears to be a need for a biosecurity assessment tool that addresses biosecurity measures in a more comprehensive way and that may be more directly targeted towards *S.* Dublin infection pathways than those already available.

This study aimed to embrace the complexity of factors affecting the probability of dairy herds becoming infected with *S.* Dublin in its approach to the assessment of biosecurity. Hence, we investigated whether based on reviewing the scientific literature, it was possible to develop a new framework with a supporting tool allowing for a time-efficient, yet comprehensive, assessment of biosecurity measures in place to prevent the introduction and establishment of *S.* Dublin in dairy herds.

## 2. Materials and Methods

### 2.1. Overview

The process of developing the biosecurity assessment framework (Figure 1) included a semi-quantitative assessment method and a supporting electronic tool (Microsoft Excel file with spreadsheets) for on-farm biosecurity assessment targeting the probability of dairy herds becoming infected with *S.* Dublin (understood as *S.* Dublin being introduced into the farm properties and spreading enough to be considered established in the herd). The framework development was organised through six phases: (i) clarification of the term biosecurity, (ii) literature review of *Salmonella* in dairy herds targeting the introduction and establishment of *S.* Dublin, (iii) a design process for the assessment method and the supporting electronic tool, (iv) inclusion of a scoring guide for the assessment method and final testing (v) expert elicitation weighting of the 12 included biosecurity assessment areas also indicated as ‘biosecurity sections’, and (vi) final adjustment.

### 2.2. Clarification of the Term ‘Biosecurity’

As a core element in the selection of relevant literature and creation of a supporting electronic tool and scoring guide, the initial process involved the identification of previous definitions of biosecurity and face-to-face discussion between the authors of this paper, all of whom have long-term experience with biosecurity, to select which definition was most useful in relation to the pathogenesis of *S.* Dublin, pathogen interaction with the host and environment, and production system features of relevance for the risk of dairy herds becoming serologically test-positive for *S.* Dublin.

### 2.3. Literature Review

An overview of the risk factors relevant to salmonellosis in calves and adult cows has been provided by other authors [39,40]. However, our review aimed at targeting the introduction and establishment of *S.* Dublin, and the literature with an outcome response of ‘*Salmonella* spp. in general or other serovars’ was critically evaluated for inclusion.

#### 2.3.1. Literature Sources and Search

An all-field free text search was conducted in three relevant databases: Ovid Agricola/Embrase, Web of Science, and PubMed. No date, geographical, or language restrictions were imposed. The search was separated into three blocks using AND, including 23 terms (Table 1). The PubMed search was supported by the Medical Subject Headings for each block. The references were uploaded to the EndNote 20 reference management tool, ClarivateTM, for removal of duplicates and manual de-duplication.

#### 2.3.2. Review including Screening and Selection of Reference Articles

The first author performed a two-step review process. The individual reference titles were first evaluated for relevance. In the second step, articles were excluded based on abstract screening if they did not appear relevant to the introduction and/or establishment of *Salmonella* spp. in cattle herds. The process was followed by full-text screening supported by the following inclusion criteria: (i) full text available in English and (ii) primary research and epidemiological investigation regarding the risk of the introduction and establishment of *Salmonella* spp. Furthermore, the selected reference was evaluated for exclusion of references not relevant regarding the introduction and establishment of the serovar Dublin in dairy herds.

### 2.4. Design of the Assessment Method and Supporting Electronic Tool for Recording of Observation and Interview Questions

Based on the literature for the review, knowledge of previously published and unpublished tools for the assessment of biosecurity pros and cons for different systems was evaluated, as well as an assessment method and framework for the supporting tool selected (Figure 1). The observation and interview biosecurity measures were selected and evaluated by the authors.

The first versions of the biosecurity assessment framework, including the supporting electronic tool and assessment method, were tested in eight dairy herds. Independently, one epidemiologist and disease modeller, a scientific expert in *S.* Dublin epidemiology (the last author of this paper), and two field experts in the prevention and control of *S.* Dublin, including the first author, tested the tool in four herds. After testing and feedback, the tool was adjusted and retested in four other herds by four veterinary experts in biosecurity and *S.* Dublin. Additionally, the interrater reliability of the initial assessment score in the observational part of the biosecurity assessment framework was evaluated in a dairy herd by nine final-year veterinary students, two research experts (including the last author), and the first author. The data from the 12 raters were statistically analysed in four categories of the scoring scale (0–24, 25–49, 50–74, and 75–100) by ‘Krippendorff’s coefficient’ [41] and as continuous data in a two-way, consistency, single rate ‘intraclass correlation coefficient’ (ICC1) [42], performed in the R statistical software version 4.1.2 (R Core Team, 2021).

### 2.5. Inclusion of the Scoring Guide

In the quantification of the biosecurity assessment, each of the chosen biosecurity sections (12 in total, Table 2) was inspired by methodologies used in the qualitative assessment of on-farm welfare and behaviour and then adapted to assess biosecurity in a holistic approach [43]. While a qualitative assessment may provide useful insights during communication with a farmer, it can be difficult to summarise for comparison between farms or if the information is to be used for quantitative analysis of factors associated with disease incursion. Consequently, to standardise the data collection according to relevance, sensitivity and specificity, robustness, feasibility, and occurrence [44], the assessment method was supported by a scoring guide resembling ethograms used in behavioural research. Scaled score definitions were inspired by the grading scale used in the Danish educational system [45]. The framework with the inclusion of the scoring guide was hereafter re-evaluated by five veterinarians with experience in control of *S.* Dublin, including the first author and a MSc. in agriculture with 20 years of experience in the construction of cattle facilities. The interrater reliability of the initial (ICC2_I) and final biosecurity score (ICC2_F) were calculated in a dairy herd, as described in Section 2.4.

### 2.6. Expert Elicitation

Additionally, each of the 12 included biosecurity sections was weighted before aggregating the overall biosecurity across all 12 sections. The process of quantifying the biosecurity was carried out by expert elicitation using a modified Delphi approach where behavioural aggregation from the four experts was obtained through virtual meetings. The weighing process was carried out in five stages: (i) the four experts were asked to independently rank the biosecurity sections from 1–12 by importance; (ii) the ranking was discussed at a virtual meeting with all four experts present; (iii) experts were asked to independently add weights to each biosecurity section, adding up to 100 points in total; (iv) a second virtual meeting was held, including a presentation of how the weighing was performed and why, followed by a group discussion of applied weights to each section; and (v) the session was closed by question adjustment and a second independent expert elicitation. The final expert weights for each of the 12 biosecurity sections were added as an overall weighted score in a separate spreadsheet to the final electronic tool.

### 2.7. Final Adjustment of the Framework

The final framework design was adjusted for easy printing and recording of scores on-farm and renamed ‘BAF-SD’, and the weighted semi-quantitative assessment method, scoring guide, and supporting electronic tool were gathered in a Microsoft Excel file with 26 named sheets.

## 3. Resulting Biosecurity Assessment Framework

### 3.1. Definition of Biosecurity

The term biosecurity has been used across different sectors, including the animal health sector. As a term, biosecurity was first cited in PubMed in 1987 [46]. Since then, a variety of definitions have come out, including the often-cited definition implemented in the European Animal Health Law regulation, the Terrestrial Code from the World Organisation for Animal Health, and different, but related, definitions in documents from the Food and Agriculture Organization of the United Nations (Table 3).

Shortened terms coexist, including ‘the combination of all the different measures implemented to reduce the risk of introduction and spread of disease agents’ [49] (p. 68) or ‘security from transmission of infectious diseases, parasites and pests’ [50] (p. 132). However, the expert group emphasizes ‘the infection becoming established’ as an essential part of the probability to detect *Salmonella* or becoming test-positive in a monitoring/surveillance programme for *S.* Dublin. This is highlighted by the ability of *S.* Dublin to survive for an extended period outside the host [51,52], and age-related susceptibility and probability and duration of shedding [53]. Additionally, cattle herds often have fewer barriers between the environment and animals compared with other livestock species kept under intensive farming conditions, such as poultry and pigs. Thus, including ‘establishment’ in the biosecurity definition is a way to nudge the biosecurity assessor to use his/her epidemiological understanding in the assessment of prevention of dairy herds becoming infected with *S.* Dublin. For the developed biosecurity assessment framework, we, therefore, defined biosecurity according to the WOAH, with a focus on the keywords ‘introduction and establishment’. Thus, we define ‘introduction and establishment’ as the combination of incidents required for a dairy herd to become test-positive in the Danish *S.* Dublin programme.

### 3.2. Search Results

The all-field free text search conducted on 14 December 2022 in the three relevant databases—Ovid Agricola/Embrase, Web of Science, and PubMed—returned a total of 3538, 3624, and 3517 documents, respectively. The automatic (in Endnote) and manual de-duplication returned a total of 5233 references. The initial two-step review process reduced the number of references to 217 and assessed eligibility for the review or the identification of other relevant studies based on the full-text screening. In total, 76 references were included in the review, and additional 38 references were identified in reference lists.

### 3.3. Review of the Probability of Introduction and Establishment of S. Dublin Infection

Disease transmission by infectious pathogens can be divided into horizontal and vertical transmission [54]. Despite the suggestions of vertical transmission of *Salmonella* [55], horizontal transmission of *Salmonella* is the main route. Experimental studies have indicated airborne transmission of *Salmonella* over short distances [56,57], and *Salmonella* spp. have been detected in aerosols from dairy calf houses and transmitted to calves [58,59]. Additionally, the highest percentage of positive *Salmonella* isolations in a 250-cow dairy herd, involving 7680 animal and environmental samples collected over a 12-month period, was from collected milking parlour air [60]. However, faecal-oral transmission through direct or indirect contact is the predominant horizontal transmission route for *Salmonella*.

*S.* Dublin is the dominating serovar in cattle in European countries [6]. Compared with other serovars that are common in cattle, such as *S.* Typhimurium, *S.* Dublin is mainly presented in low numbers in the environment and manure in infected herds [61]. However, *Salmonella* bacteria have the ability to survive in slurry for months and even years in dried manure [51,52].

#### 3.3.1. Direct Contact

*S.* Dublin is a host-adapted serotype with a possible carrier state [62], and direct contact between cattle through purchasing, common grassing, and mingling either due to grassing or heifer raising off-farm is acknowledged as an important risk factor for between-herd transmission of *S.* Dublin [30,63,64,65] and *Salmonella* more generally [66,67,68]. In a few studies, a negative association between the presence of *S.* Dublin and purchases has been observed. This may be due to introduction of recall bias due to the questionnaire study design or, as suggested by Ågren et al., 2016, a low prevalence in the population [33,69]. Surprisingly, common pastures were not associated with the presence of *Salmonella* in all studies. In a study of *Salmonella* incidence in dairy herds in the United Kingdom, common pastures and/or rented cattle grassing areas protected against the introduction of *Salmonella*, and this was supported by a Dutch case-control study for the identification of *S.* Dublin risk factors in dairy herds [63,70]. None of those studies explained the negative association. A suggested explanation may include a lower density of cattle at pasture than during indoor housing, and a lower probability of contact with contaminated housing facilities [71]. Furthermore, the selection criteria for *S.* Dublin herds in the Dutch case-control study included laboratory findings, which may have challenged the sensitivity and specificity of the included study units. However, transmission through common pastures is supported by epidemiological investigation and genome analysis [72] and may be due to indirect transmission in pasture areas or surroundings [37].

#### 3.3.2. Indirect Transmission Pathways

##### Pasture Vehicles (Slurry, Cowpats, Water)

*S.* Dublin can survive for weeks on the grass leaves after the application of contaminated slurry on pastures, and transmission of the pathogen to grassing animals has been documented [73,74]. In both these studies, the inoculated slurry contained low dry-matter content (<0.9%), leading to a potentially shorter decimation time [75]. However, the influence of dry matter on the *Salmonella* survival period in the slurry is inconsistent. For example, in a field study including dairy cattle slurry inoculated with *S.* Typhimurium, the effect of dry matter on pathogen survival was not consistent [76]. However, it is reasonable to believe that a higher dry matter content will extend the survival period because of less exposure to UV radiation. This is supported by the ability of *S.* Typhimurium to be viable in cow pats for up to a year, depending on the decomposition influence by season and weather conditions [77]. Most slurries in Denmark applied to grassing areas or areas used for forage have a dry matter content of >5% and are often injected into the soil. The study results of one month of pathogen survival after injection in the soil by Nicholson et al. is, therefore, representative of the transmission risk [76]. Furthermore, cow pats may act as a reservoir for *Salmonella* transmission between grassing seasons [76]. Pasture areas used by other foreign cows, application of manure from infected herds close to or directly on pastures without an adequate resting period is, therefore, thought to be an important *S.* Dublin transmission route and an efficient pathway for the establishment of the bacteria within a herd [35]. The probability of establishment was highlighted by Fossler et al., who identified a significant risk of shedding of *Salmonella* bacteria in cows in dairy herds with previous *Salmonella* history if their own manure was applied to owned or rented land or if cows ate roughage or grassed fields where manure had been applied to the ground surface without additional ploughing [78]. Of the related factors associated with the introduction of *Salmonella,* Hughes et al. believed that most of the incidence of *S.* Dublin over a 12-year period in South Wales was due to contamination of water courses either by grassing animals or by effluent [79]. To the best of our knowledge, this hypothesis has never been demonstrated scientifically for *S.* Dublin, but in a previously referred Dutch case-control study, the total area of the water surface was positively associated with *S.* Dublin infection [63] and *S.* Dublin has been isolated from rivers close to grassing cattle in North Wales [80]. Additionally, a Swedish epidemiological investigation of *S.* Dublin-infected cattle herds involving whole genome sequencing identified multiple links between clustered farms and potentially contaminated drinking water due to common water course access by foreign cattle [37]. This association was also identified for *Salmonella* in an Irish study involving 309 dairy herds. The odds ratio for the presence of *Salmonella* antibodies in bulk milk was 5.3 (95% CI 1.34, 20.97) times more likely if cattle frequently had access to watercourses passing other farms compared with no access [68].

##### Entrances and Visitors

The structure of dairy herds often requires visits of persons and vehicles with contact with other farm environments (e.g., contractors, biogas trucks, livestock transport, feed company trucks, rendering company trucks, veterinarians, artificial insemination technicians, hoof trimmers, and milk collection trucks). This frequency typically increases with herd size, as indicated in a Swedish study that logged visitors over periods of four weeks, and where the daily average number of professional visitors and visitors in direct contact with animals was correlated to herd size [81], thus, increasing the accumulated risk of disease transmission, As an example, there is a tendency towards an association between frequent visits of hoof trimmer (*p* = 0.07) and the presence of *Salmonella* antibodies in BTM in endemic areas [82].

Depending on the farm structure, facilities, and management, visitors and vehicles may get close to internal driveways and sometimes even get in contact with animals or their feed [81]. The risk of transmission via persons and vehicles is, therefore, herd-specific and may interact with the seasonal pattern of *Salmonella* introduction, with a higher risk in autumn, where higher temperatures, rainy weather, muddy driveways, and entrance areas favour initial bacterial growth and survival before decimating over time [83]. This was supported by a longitudinal study based on 19,296 environmental samples from 449 dairy farms in England and Wales. In that study, the lack of clean visitor parking areas (as a proxy for on-farm hygiene) was found associated with an increased incidence of *Salmonella* [70]. Despite its herd specificity, personal clothing and transport ranked as posing a relatively high risk of disease transmission [49,84].

Owing to the close contact with animals and their environment by professional visitors in dairy herds, compromised personal hygiene or cleanliness of equipment may result in the transmission of *S.* Dublin. *Salmonella* spp. have been shown to survive on thoroughly rinsed booths stored for 48 h after use in cattle herds [85]. However, strong evidence of the influence of personal hygiene on herd transmission of *S.* Dublin is lacking. Nonetheless, veterinarians have been used as a proxy for professional entrance in the probability of disease introduction. Among the 95 Dutch dairy herds included in a longitudinal cohort study, disease outbreaks with BHV 1, BVDV, *Leptospira hardjo*, and *S.* Dublin could to some extent be prevented by the everyday use of protective clothing by visitors [65]. Additionally, cow operations require heavy workloads at odd times and often involve part-time workers. Davison et al. argued that the risk of dairy herds becoming *Salmonella*-infected increases when there is at least one part-time worker and may be linked to part-time employment on several farms [70].

##### Sharing of Equipment and Transportation of Manure

Farmers frequently share farm equipment. In 56 cattle farms in northwest England, 43% of farmers responded that they shared equipment with other farmers [86], and sharing of machinery was also suspected to be the cause of *S.* Typhimurium DT104 transmission between three cattle herds in a Danish outbreak investigation in an endemic area [87]. On the other hand, sharing equipment used for manure handling in 281 of 442 Swedish dairy farms was not found to be significantly associated with the occurrence of *S.* Dublin antibodies in bulk tank milk [88]. However, the equipment can be shared with different frequencies and under different conditions, and the question may have been too general to be a predictor for (often delayed) *Salmonella* antibody reactions in bulk milk in that study under low prevalence conditions in Sweden. Indeed, whole-genome sequencing and epidemiological investigation identified the sharing of equipment used for manure and feed handling as a possible link between two clustered *S.* Dublin outbreak herds in the same country [37].

Most modern dairy farms store manure as slurry in slurry tanks and as solid manure directly in stacks, unless the manure is delivered directly for biogas production or field application. *Salmonella* bacteria in slurry and solid manure decimate over time, but the process is influenced by many factors, including pH, number of bacteria, dry matter content, aerobic condition, and particularly temperature [75,89,90]. Thus, *Salmonella* appears to die out faster in solid manure than in slurry [76,91], and increasing temperatures can reduce the concentrations of *S.* Dublin and other pathogens in biogas plants, which has been suggested as a valuable benefit of biogas production [92]. However, the on-farm location of slurry tanks and solid manure stacks often enables cross-contamination of traffic, feed, or cattle, and the purchase of manure and sharing of slurry tanks between herds has been associated with an outbreak of *S.* Typhimurium and identified as an epidemiological link between *S.* Typhimurium DT 104 outbreak herds, respectively [71,87]. Additionally, delivery to biogas plants leads to a high number of extra incoming transport of digested residues and outgoing transport of manure that often encounter or cross internal driveways because fresh slurry and deep bedding materials are often collected near the barn. Furthermore, slurry trucks are not always cleaned between every transport, and the inside of the tank may be able to transfer small amounts of residual non-treated manure between farms. Such contamination was indicated in a study involving four commercial biogas plants, where digested residues in farm wells of different origins contained *Salmonella* Agona with the same pulsed-field gel electrophoresis pattern at two farm sites [93]. Consequently, with an average yearly production of 35 to 45 tons of slurry per Holstein cow and replacement heifer, the yearly accumulated probability of disease transmission resulting from manure handling and transportation is substantial, even at a low probability for each event. For instance, the yearly number of manure transport can exceed 500 in a 250-cow dairy herd.

##### Transportation of Livestock

Shipping of livestock may stress animals, and sampling of cattle faeces from the rectum and swabs of hides before and after short shipping identified a 2- to 14-fold increase in the prevalence of positive samples for *Salmonella* spp. [94]. In the same study, 74.5% of the livestock transport vehicles were positive for *Salmonella* spp., suggesting that transportation vehicles are critical control points in biosecurity assessments. According to the Animal Health Law, livestock transport vehicles are required to be cleaned and disinfected after animals are unloaded [95]. However, this may not always be accomplished (or accomplished effectively), and sampling of vehicles used for the transportation of calves to markets identified *Salmonella* spp. in 20.6% of examined transport at unloading but before washing [96]. After washing, 4 out of 62 (6.5%) vehicles were positive for *Salmonella* spp., and *S.* Dublin was isolated from 3 out of 5 positive samples [96]. It should be noted that performing a study may affect procedures, and hygiene may have improved and, thereby, may introduce bias in the results. Yet, it highlights contaminated vehicles as a potential risk factor for disease transmission. Indeed, preliminary results from an ongoing Spanish survey indicate failure in the cleaning procedures of livestock transporters [97]. Inadequate washing and disinfection were observed in 26% of the inspected livestock transporters for cattle by the Danish authorities during 2020 and 2021; pickup of animals is often carried out near the barn without the required hygiene barriers, indicating that precautions regarding biosecurity at on-farm animal loading facilities and cleaning and disinfection of cattle transport are not always adequate [81,98]. The transportation transmission pathway is also hypothesized to be the reason for the introduction of *S.* Dublin to the state of New York, Pennsylvania, and Ohio back in 1988 [99].

##### Carcass Disposal (Rendering Collection Point)

Rendering trucks are considered a potential risk for disease transmission, and storage places of carcasses are recommended to be located close to a public road to prevent the crossing of on-farm driveways [100]. To the best of our knowledge, carcass disposal has not been studied for its association with herd transmission of *Salmonella* spp. However, scavengers, such as red foxes, have been suggested as a reservoir for *S.* Dublin due to the demonstration of this serovar in the liver and intestines of five (1.2% of 434 tested) foxes in Tyrol [101]. Pools of faeces from foxes collected from pasture areas close to chamois carcasses infected with *S.* Dublin also tested positive for *S.* Dublin [101]. Thus, the shielding of carcasses from scavengers should help prevent the introduction of *Salmonella* into a farm.

##### Pets, Birds, and Vermin

*S.* Dublin has been isolated sporadically from animals other than cattle [79,101,102,103,104,105,106,107,108], but clinical disease is mostly reported in cattle, humans, and mink. However, other species have also been suggested as reservoirs or passive carriers. For instance, a herding dog tested bacteriologically *S.* Dublin positive weeks apart in endemic areas of Southern Bavaria, Germany [72]. Rodents and wild birds are reservoirs for *Salmonella* spp., but only a few studies have isolated *S.* Dublin from rodents in nature and wild birds [104,109,110,111,112]. In a study in 36 Polish cattle herds, 1124 house sparrows were captured and bacteriologically cultured from the liver, heart, intestine, and surface, including beak, wing tips, and legs. Twenty samples from the heart, liver, and intestines (from an unreported number of birds) were positive for *S.* Dublin. These findings suggest a potential carrier state for house sparrows, although the number of positive individuals was not stated [109]. In general, the prevalence of *Salmonella* in wild birds observed on agricultural land and cattle farms is low, but most studies have been carried out in countries where serotypes other than *S.* Dublin dominate in the cattle population and potentially in the wild birds [6,113,114,115,116]. Additionally, numerous birds can dispose of high concentrations of bird faeces in open storage of feed, and a low prevalence may accumulate to a potential source of *Salmonella* infection [117]. Indeed, numerous birds may mechanically transmit *S.* Dublin bacteria between farms.

##### Feed

Although housed cattle are often fed heat-treated pelleted concentrate and ensiled roughage, limiting the survival of *Salmonella* bacteria [118], relatively apathogenic or non-host-adapted *Salmonella* serovars, but rarely *S.* Dublin, may be isolated from feed [106,107,119,120]. Among others, *Salmonella* culturing of 4582 pooled feed samples from 59 dairy herds in Washington State identified feed positive for *Salmonella* in 12 herds. However, only one positive *S.* Dublin isolate was gathered sterile from calf grain despite that *S.* Dublin was the third most frequently isolated serotype among the 56% of herds with at least one positive faecal sample out of 7009 pools. Adhikari et al. hypothesised that feed contamination may have occurred on farms, due to a low number of positive *Salmonella* samples at feed mills (1 out of 665 samples) [119]. The *S.* Dublin-positive calf grain was collected from a herd without *S.* Dublin-positive faecal samples, and *S.* Dublin was isolated from feed mills in a previous study [106]. Thus, *S.* Dublin introduction through feed cannot be ruled out. Feed is frequently stored in open areas that may be located close to external transport routes or washing facilities. Therefore, feed is at risk of being contaminated. In univariable analyses, open storage of silage and concentrate was found to be associated with dairy farms being positive for *Salmonella* spp. in England and Wales [70]. Modern feeding procedures often involve feeding a total mixed ration (TMR) and spot contamination of feed, thereby posing a high risk of disease establishment because the pathogen can be introduced to many animals upon mixing. A collection of environmental samples from 18 different cattle farms in the United States identified *Salmonella* occurrence in 12.5% of samples from fresh TMR [121]. In addition, an epidemiological investigation of an *S.* Typhimurium outbreak in an endemic area of Denmark found a link between two clonally similar outbreak herds, where feed was brought in from one infected herd to the other [87]. In a similar investigation of *S.* Dublin in Sweden, a wagon used for feed was an identified link between two clustered herds [37].

#### 3.3.3. Establishment of Infection

When considering the probability of the establishment of introduced *Salmonella* bacteria, the segregation of animals is an important preventive measure. This is supported by the higher risk of introduction and longer recovery period in larger and more complex herds in case of *Salmonella* infection [30,122,123] and, also, by the lower risk of cows shedding *Salmonella* in tie stalls compared with loose housing systems [78,124]. Indeed, the hazard ratio for recovery in large Danish dairy herds was among the lowest across investigated predictors over a 10-year period [122]. Furthermore, segregation is vital in the housing and handling of more susceptible and infectious animals. Especially calves are highly susceptible and infectious [53,125,126]; also, transition cows, overstocked cattle, and sick animals are typically more susceptible and infectious [127,128]. Indeed, a period without calving strongly reduces the risk of clinical outbreaks of *Salmonella,* and the probability of detecting *Salmonella* in calves has been shown to be lower in herds with good than poor calving and calf management, including a maximum of four cows in a calving pen, fewer persons responsible for calving and colostrum handling, and not using calving pens for sick animals [71,129,130]. Losinger et al. found a reduced risk of *Salmonella* shedding in pre-weaned calves from dairy farms with individual animal areas for calving [131]. Furthermore, feeding colostrum from the calf’s own dam has been associated with preventing of *Salmonella* clinical outbreaks [71]. This may reflect a risk of bacteriological cross-contamination at the pooling of milk and *Salmonella* bacteria’s ability to survive in milk and even grow under favourable conditions [132,133,134]. Still, in one study contamination of milk with *Salmonella* was not considered important in the transmission of the pathogen to offspring [135]. Segregation of pre-weaned calf pens by solid walls was also associated with the prevention of *Salmonella* exposure [129]. This may explain why organic production with requirements to extend cow-calf contact time and the requirement of common pens for pre-weaned calves has been associated with a higher risk of becoming or remaining test-positive [64,122]. However, Hardman et al. found a tendency towards shorter time to infection with *Salmonella* in individually penned calves at longer distances from index infected calves, suggesting the importance of biosecurity measures regarding indirect transmission, including enhanced hygiene level and reduced risk of aerosol production [136]. Indeed, the hygiene of cow stables or calf facilities has been associated with the presence of *Salmonella* in dairy herds [67,82,137] and may be the reason dairy herds with increased somatic cell counts are associated with the risk of becoming test-positive for *S.* Dublin [30]. Good hygiene is not only relevant in animal facilities; reducing the risk of contamination of feed storage areas has also been associated with a reduced risk of *Salmonella* shedding in cows [78].

Other factors may also enhance susceptibility or excretion of *Salmonella* and, thereby, increase the probability of establishing the infection within the herd, including metabolic or heat stress, other diseases, etc. Liver flukes are associated with the occurrence of *S.* Dublin in dairy farms [63], and cattle infected with *Fasciola hepatica* have been found to have increased susceptibility to experimental intravenous infection with *S.* Dublin and persistent excretion of the bacterium at higher numbers and frequencies [138,139]. It has been difficult to replicate these results by oral administration of *S.* Dublin [140]. However, exposure to sublethal oral doses of *S.* Dublin in fluke-infected cattle may enhance the likelihood of persistent excretion of *Salmonella* compared with non-fluke-infected cattle [141].

All of the above reviews of critical risk factors and control points in hygiene and biosecurity measures were used to guide the development of the new biosecurity assessment framework, including both which sections to include and which questions and points to address within each section and finally to guide the weighing of the sections by the experts.

### 3.4. Biosecurity Assessment Framework for S. Dublin (BAF-SD)

The final biosecurity assessment framework ‘BAF-SD’ is provided in English in Appendix A, subdivided into 26 printing-optimised spreadsheets including the ‘*user guide*’, the ‘*scoring guide*’, the ‘*supporting electronic tool*’ and the ‘*summary spreadsheet with weighted assessments*’. The 12 biosecurity sections each include an observation section, an interview section, and three boxes with the ability to assess the initial, the adjusted, and the final biosecurity assessment.

BAF-SD is used to score biosecurity practices on farms in seven steps as illustrated in Figure 2. In the first step (~10 min), a preliminary quick check is performed with the farmer/herd manager at the farm entrance to obtain an overview of the farm area and barn sections. This includes 10 demographic/background questions and a drawn farm overview. A map, for example, as a GIS orthophoto of the farm at 1:1000, 1:2500, or 1:5000, which is downloaded to gather location information, logistics of transportation vehicles entering, internal transport, crossing driveways, and borders between external and internal contacts (Figure 3).

In steps 2–3 (~50 min), the biosecurity assessor walks between the biosecurity sections alone and performs an independent initial biosecurity assessment based on his/her first impressions. The principle of the assessment is to score each biosecurity section on a scale from 0 to 100. Score 0 is the worst performance with essentially no biosecurity measures in place for a given section, and 100 is the maximum score given for a high level of command of all aspects of biosecurity for a given section, with no or only a few minor weaknesses. If the biosecurity level for a section does not meet the specified minimum requirements assessed by the biosecurity assessor, the section should be scored as 49 or below. The initial biosecurity assessment is followed by filling in the observation checklist and repeating the protocol for each of the 12 biosecurity sections. The observation checklist considers a total of 56 variables on an ordinal scale.

In steps 4–5, the farmer and biosecurity assessor revisit the 12 biosecurity sections by a farm walk together and re-evaluate whether there were missing or incorrect observations according to information from the owner/manager. The observational checklist is adjusted, and a second biosecurity assessment based on the impression and gathered information in the observation checklist is filled in (adjusted biosecurity assessment).

The final steps 6–7 involve an in-depth interview using a questionnaire-interview checklist. Ideally, the interview should be carried out by asking open questions to allow the farmers to describe their procedures as performed, and the biosecurity assessor then gathers the answers into selective questions on a dichotomous or ordinal scale based on the conversation with the farmer. Any missing scores must be revisited by addressing specific semi-closed questions as indicated in the questionnaire. Finally, the biosecurity assessor fills in the final biosecurity assessment based on all impressions, gathered information, and the ‘*scoring guide*’. This process is repeated for each of the 12 biosecurity sections. In total, 109 interview questions are addressed across the biosecurity sections referring to the biosecurity measures used on the farm within the last 12 months. Steps 4–7 can be conducted within a timeframe of ~1 h.

### 3.5. Interrater Reliability

The first validation was conducted in a closed 334-cow dairy herd with no animals in the pasture. The second validation was conducted in 748-cow dairy herds with heifers raised at another location, animals in the pasture, and the purchase of animals within the previous year. The 12 initial biosecurity assessment scores indicated a variety of agreements depending on the biosecurity assessment area (results not shown) and an overall poor correlation for the four grouped ordinal scales by the Krippendorff’s coefficient; α = 0.205 and α = 0.0919, respectively. Additionally, the continuous variable by the intraclass correlation coefficient for the initial biosecurity assessment was ICC 1_I = 0.37 and ICC 2_I = 0.19, respectively. The interrater reliability improved for the final biosecurity assessments carried out in the second validation by adding interview questions and the scoring guide, returning a Krippendorff’s coefficient α = 0.442 for the grouped ordinal scale and the interclass correlation coefficient (ICC 2_F) = 0.60 (Table 4).

### 3.6. Resulting Weights of Biosecurity Sections

The central question to be weighted for each biosecurity section = X (Table 2), adding up to 100 points in total was: “Given new infection with S. Dublin expressed as recently becoming serologically positive on bulk tank milk, what is the probability that the introduction and establishment of the disease has occurred through biosecurity section X out of the 12 selected biosecurity sections, given the other biosecurity sections represent a typical Danish dairy herd”, returning final resulting mean weights in Table 5.

## 4. Discussion

The main objective of the described work was to develop a biosecurity assessment framework that includes a supporting electronic tool. This was achieved, and the BAF-SD tool is made freely available in English as Appendix A with this paper. This makes it feasible to assess on-farm biosecurity for further research-based knowledge about the indirect spread mechanisms and drivers for the local spread of *S.* Dublin infection between dairy herds. Furthermore, the tool is most likely relevant to other pathogenic *Salmonella enterica* serovars and other pathogens spread by animals and manure between dairy cattle farms. Focusing on preventing *S.* Dublin infection in dairy herds, we highlighted terms such as ‘introduction and establishment’ in the biosecurity definition and downplayed terms such as ‘spread and persistence’ not only for the use of the framework but also for the critical selection of the literature.

A literature search regarding risk factors for the introduction and establishment of *S.* Dublin in dairy herds identified some knowledge gaps. The main issues with the available literature were as follows: (i) The first issue was the failure to identify single environmental risk factors that would be useful to target the prevention of the introduction and establishment of *S.* Dublin in dairy cattle farms. This is likely due to the complexity or multitude of the *S.* Dublin introduction pathways combined with a relatively low number of published studies, mostly based on small sample sizes. The difficulties in identifying single risk factors for the *S.* Dublin introduction and establishment in dairy herds have also been described for *Salmonella* by others [88], and requests for more general biosecurity assessment tools have been posed. Thus, the value of identifying a single factor can be discussed, as it does not represent the real-life transmission pathways on dairy farms. To the best of our knowledge, only one published assessment tool addressing overall biosecurity in dairy herds is available, i.e., Biocheck.UGent, which is available on the website https://biocheckgent.com/en (accessed on 11 August 2023) and has been scientifically described [28]. However, the goal was to be able to assess the multiple introduction and establishment pathways of *S.* Dublin. We decided to achieve this through a semi-quantitative approach focusing on pathways relevant to the transmission of bacteria through manure and water-splashing events and where the biosecurity sections were weighed with S. Dublin in mind. We, therefore, created a new tool. (ii) The second issue was case reports or small studies that were not able to refute their null hypothesis. Nonetheless, several case reports and studies suggesting herd links based on bacteriological identification, genome analysis, and/or epidemiological investigation were included with precaution. (iii) The third issue was the low number of *S.* Dublin-specific studies, so other studies of other *Salmonella enterica* serovars were critically reviewed and included if they were found to be appropriate. This must be taken into consideration as the transmission pathways, spread patterns, and concentrations of bacterial excretion differ somewhat between different serovars. (iv) The fourth issue was varying accuracy in test schemes and selection of study units introducing possible misclassification bias. *Salmonella* shedding is intermittent [142], and *S.* Dublin bacteria are often excreted and isolated in lower percentages from animals and the environment compared with other serovars, such as *S.* Typhimurium [61]. Therefore, the sensitivity of the bacteriological culture of animals is as low as 6% in asymptomatically infected cattle [143]. In comparison, a high (>95%) herd sensitivity and specificity can be achieved by including repeated bulk tank milk (BTM) and even higher by including serological testing of specific animals [144,145]. Therefore, the selection of study units based on previously reported *Salmonella* outbreak(s), detection of the bacteria by bacteriological faecal culture, and indirect ELISA test on serology and BTM test scheme add different levels of study quality. (v) This brings us to the next issue—the studies were mostly carried out as questionnaires designed with accuracy and precision issues. Self-reporting bias is not unlikely for questionnaire studies, which has been observed in self-reporting biosecurity programmes in an online setting compared with results obtained during consecutive farm visits [146]. Thus, to develop a successful and reliable biosecurity assessment framework, it is important to combine farm observations and interviews that can be easily conducted by a trained biosecurity assessor. To be successful, an assessment tool also needs to be completed within a limited timeframe; therefore, the complexity and number of questions are with limitations. This challenges the ability to perform a quantitative assessment covering all aspects of farm biosecurity, which often is farm specific and complex. The qualitative assessment adds the ability to include different aspects of observations and different questions with the purpose of better understanding the subject, for example, prevention of the disease introduction, but with less rigour in data collection [147]. Such an observational approach has been used in behavioural expressions in pigs with significant inter-observer agreement levels [148]; therefore, it was decided to include gathered quantitative observations and interview information in a semi-quantitative assessment approach to strengthen the validity and reliability of the framework.

Interclass correlation of assessing observed biosecurity with minimal information (initiated biosecurity assessment) had an ICC estimate below 0.50 (Table 3), indicating poor reliability. Consequently, a ‘*scoring guide*’ was included, returning an ICC estimate of 0.60 (95% CI 0.37, 0.83), indicating improvement and moderate reliability [42]. The scoring guide for each biosecurity section is useful in the training and calibration of biosecurity assessors. It can also support the assessors during the scoring on the farm but is probably too comprehensive to guide the whole process on the farm. Therefore, training and good preparation for the farm visit are needed. Furthermore, the initial assessment indicated different perceptions of biosecurity between raters regardless of their education level and field experience, so scoring guides are necessary to standardise the assessment of biosecurity. However, the reliability of the results must be interpreted with caution. The lower boundary for the 95% confidence interval is below 50, and the upper boundary is above 75; in the worst and best cases, the reliability of the full framework is poor and good, respectively. Indeed, the final assessment reliability test was carried out in only one herd by six raters. A complementary reliability test should include at least 3 raters and 30 heterogeneous observations, and further testing is required [42]. However, the specific tool was designed for hypothesis-generating research studies, to gain more knowledge about the risk of *S.* Dublin’s introduction in a complex setting. The tool targets use by the same or a group of trained assessors. Therefore, intra-observer reliability can be evaluated to some extent by comparing the information gathered from the observation and interview scheme with the spontaneously assessed score of each section. In both inter-observer reliability tests, raters were not trained or calibrated, and this may also be part of the reason for the low to moderate interrater agreements.

The expert elicitation was conducted by the four authors, due to their scientific experience in *Salmonella*, biosecurity, epidemiology and disease control, and in-depth knowledge of the designed biosecurity assessment framework. As illustrated in Table 4, the ranking and weight of biosecurity sections varied among the experts. The process identified central obstacles in the ranking and weight of biosecurity sections, and despite discussions between elicitations, a complete consensus was not reached. During the discussion, some of these obstacles included the following: (a) The first obstacle was the level of complexity due to the inclusion of both introduction and establishment in the assessment framework. As expressed by an expert, the risk of the introduction for the biosecurity section ‘Animals on pasture’ was considered high, but that of the establishment, low. (b) The second was legalisation enforcement. By legalisation, visits to test-positive herds should be limited and should be placed after visiting test-negative *S.* Dublin herds. However, whether this rule is fully followed has never been investigated, and the knowledge could have elicited another distribution of weights. (c) Another obstacle was the interactions between the biosecurity sections. For instance, storage of feed, solid manure, and washing facilities may be located at the same spot on some farms, at a short distance from animals. In such cases, the effects of poor hygiene and lack of transmission barriers would overlap between the sections. Thus, it can be argued whether the low number of recommended experts (8–20) and the low correlation of opinions about the ranking and weights of the sections represent the real-life average situation [149,150].

## 5. Conclusions

The developed biosecurity assessment framework provides a novel approach to comprehensively assess the overall biosecurity level of relevance of *S.* Dublin introduction and establishment through a structured approach addressing 12 specific dairy farm sections within a timeframe of approximately 2 h. The scoring is based on on-farm observations and interviews. This can provide a useful tool for researchers and herd health consultants by facilitating the generation of new and systematically collected information about biosecurity on dairy farms and will be useful in future epidemiological studies on risk factors for the introduction and establishment of *S.* Dublin in dairy farms. Importantly we have integrated the environmental factors, housing facilities, and logistics into the framework.

## Figures and Tables

**Figure 1 animals-13-02649-f001:**
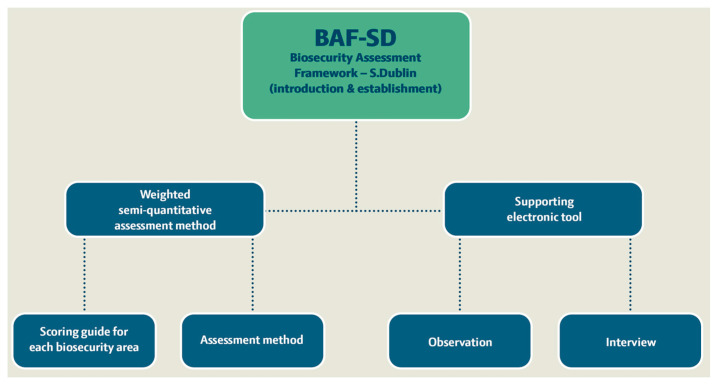
Overview of the elements of the new biosecurity assessment framework for *S.* Dublin introduction and establishment on dairy cattle farms (BAF-SD).

**Figure 2 animals-13-02649-f002:**
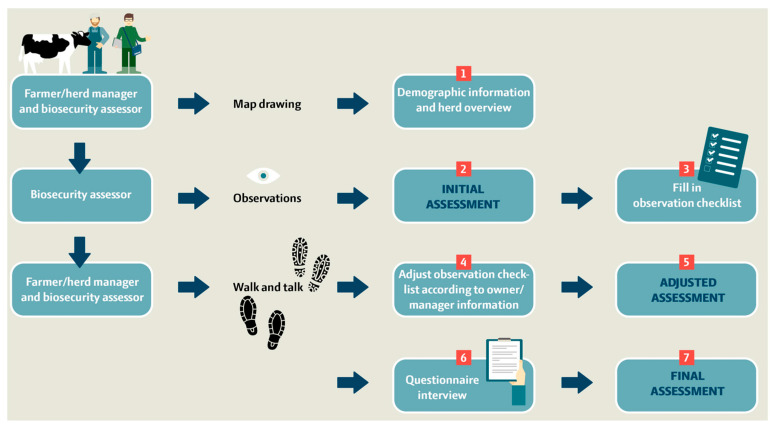
Flowchart of the 7-step process for performing on-farm observation and interview-based biosecurity scoring and data collection using the *S.* Dublin-targeted biosecurity assessment framework (BAF-SD) for dairy cattle farms.

**Figure 3 animals-13-02649-f003:**
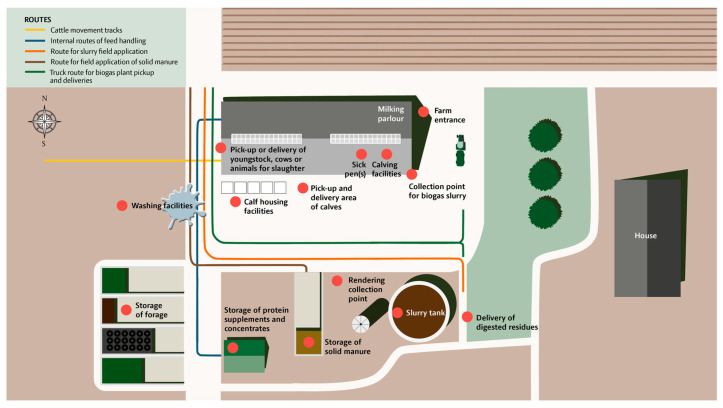
Illustration of a farm overview. Logistics include routes of movement of livestock manure, internal routes of feed handling, and possible cattle movement tracks. Fourteen predetermined locations are marked on the map, including locations for farm entrance, pickup-delivery of youngstock, cows or animals for slaughter, calving facilities, sick pen, calf housing facilities, slurry tanks, storage of solid manure, a collection point for biogas, delivery of digested residues (biomass), storage of protein supplements and concentrates, storage of forage, washing facilities, and rendering collection points.

**Table 1 animals-13-02649-t001:** Search profile used for overview identification of the literature evaluating introduction and establishment of *Salmonella* in Ovid Agricola/Embrase, Web of Science and PubMed on 14 December 2022. PubMed search was supported by the Medical Subject Headings (MeSH). Asterisk (*) represent truncation.

No.	MeSH (PubMed)	Search Terms
1.	“*Salmonella*”, “*Salmonella* Infections”, “Animal”	“salmonell*”
AND		
2.	“Cattle” “Cattle Diseases”	“cattle*” OR “bovin*” OR “cow” OR “cows” OR “calf” OR “calves” OR “heifer*”
AND		
3.	“Disease Transmission”, “Infectious”	“introduct*” OR “transport*” OR “spread*” OR “feed*” OR “silage” OR “transmiss*” OR “transmit*” OR “contamin*” OR “carriage” OR “carrier*” OR “bird*” OR “rodent*” OR “flies” OR “wildlife” OR “infect*”

**Table 2 animals-13-02649-t002:** Biosecurity sections included in the Biosecurity Assessment Framework for *S.* Dublin introduction and establishment on dairy cattle farms (BAF-SD).

Number	Biosecurity Section	Number	Biosecurity Section
1	Entrance	7	Manure
2	Pickup-delivery of calves	8	Storage of feed and feeding
3	Pickup-delivery of adult cows	9	Washing facilities
4	Calving facilities	10	Animals on pasture
5	Calves < 130 days old	11	Vermin control
6	Cattle older than 130 days	12	Carcass disposal

**Table 3 animals-13-02649-t003:** Example of alternative definitions for biosecurity. * Several definitions have been used by FAO.

No.	Definition for Biosecurity	Reference
1	‘…the sum of management and physical measures designed to reduce the risk of the introduction, development and spread of diseases to, from and within: (a) an animal population, or (b) an establishment, zone, compartment, means of transport or any other facilities, premises or location…’	The European Animal Health Law Regulation [1] (*Chapter* 1 *Article* 4 nr. 23)
2	‘…a strategic and integrated approach that encompasses the policy and regulatory frameworks [including instruments and activities] that analyse and manage risks in the sectors of food safety; animal life and health; plant life and health, including associated environmental risks.’	The Food and Agriculture Organization of the United Nations (FAO) * [47] (p. 61)
3	‘… a set of management and physical measures designed to reduce the risk of introduction, establishment and spread of animal diseases, infections or infestations to, from and within an animal population.’	World Organisation for Animal Health (WOAH) [48] (p. 2)

**Table 4 animals-13-02649-t004:** Interrater reliability by intraclass correlation for initial biosecurity scores by two groups of observers (raters) for two different dairy herds, and for a second assessment, reliability after inclusion of an interview and use of the scoring guide in the final biosecurity assessment.

		Raters	Intraclass Correlation	95% Confidence Interval	
				Lower Bound	Upper Bound
ICC 1_I	Initiated biosecurity score	12	0.37	0.17	0.68
ICC 2_I	Initiated biosecurity score	6	0.19	0.011	0.51
ICC 2_F	Final biosecurity score	6	0.60	0.37	0.83

**Table 5 animals-13-02649-t005:** Results of expert elicitation with ranks and final mean weights per expert and overall for each biosecurity section included in the BAF-SD.

No	Biosecurity Section	Expert								Mean Weights
		1		2		3		4		
		Rank	Weight	Rank	Weight	Rank	Weight	Rank	Weight	
1	Entrance	4	10	5	10	2	14	6	8	0.10500
2	Pickup-delivery of calves	3	12	3	12	3	13	9	5	0.10500
3	Pickup-delivery of adult cows	5	8	8	6	4	11	11	2	0.06750
4	Calving facilities	7	8	7	7	8	7	4	10	0.08000
5	Calves < 130 days old	8	8	6	8	9	6	1	21	0.10750
6	Cattle older than 130 days	9	7	9	5	10	5	7	8	0.06250
7	Manure	1	18.5	1	20	1	15	2	16	0.17375
8	Storage of feed and feeding	2	14	2	16	6	8	5	9	0.11750
9	Washing facilities	10	5	10	2	7	8	3	10	0.06250
10	Animals on pasture	6	8	4	12	5	9	8	6	0.08750
11	Vermin control	12	0.5	11	1	11	2	12	2	0.01375
12	Carcass disposal	11	1	12	1	12	2	10	3	0.01750
			100		100		100		100	1.00000

## Data Availability

Data sharing is not applicable, as no new data (apart from validation data on the reliability of scoring between observers) were created or analysed in this study.

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
