# Peer review of "Semi-Quantitative Biosecurity Assessment Framework Targeting Prevention of the Introduction and Establishment of Salmonella Dublin in Dairy Cattle Herds"

_animals, 2023, doi:10.3390/ani13162649_

Round 1

Reviewer 1 Report

In this study authors present an approach based on existing knowledge of biosecurity, useful for evaluating the risk of dairy herds becoming infected by cattle-adapted diseases like Salmonella Dublin. The resulting biosecurity assessment framework includes a weighted semi-quantitative assessment method with a scoring guide, in an electronic supporting tool, for 12 biosecurity sections assessed through on-farm observations and farmer interviews. I think that this work advances the current knowledge on biosecurity assessment because, through the BAF-SD 593 tool,  it provides a time-efficient and comprehensive evaluation of biosecurity measures taken on-farm and it helps the FBO to identify, categorize and minimize existing risks on dairy farms. In addition, much appreciated the choice to made the tool freely available in English to trained biosecurity assessors (you can find it as supplementar material attached to this manuscript). If compared with other biosecurity assessment tools, the BAF-SD 593 is based on interviews of FBOs not only on observations of rankers and because of this, it seems like a more comprehensive tool.
The involvement of the personnel on farms justifies the presence of “Institutional Review Board Statement” and the “Informed consent Statement” at the end of the manuscript.
The manuscript is scientifically sound. In addition, the manuscript’s results are reproducible based on the details given in the methods section that is very detailed and accurate, especially the research and selection of literary sources.
For easy reading, authors could:

Line 107: Add the expression “also indicated as “biosecurity sections” after “biosecurity assessment areas”.

Combine subparagraphs 3.3.2.2. e 3.3.2.2.1.

Authors should refer to Salmonella spp. in a unique and proper way within the text (es. line 140; line 143; Table 1 n.1).

Line 408: Please revise the format of the title.

Line 388: Please add the reference after “samples for Salmonella spp.”

Lines 392-395: For the sake of clarity, please specify the study you are referring to both in the sentence and in the brackets before the dot.

Line 398: Please add the reference to the sentence.

Line 408: Maybe authors should change the title of this subparagraph in “Carcass removal” in line with Table 2 or viceversa.

Line 415: Please add the reference to the sentence after “Tyrol”

Line 417: Please add “into a farm” after “Salmonella”.

Tables and figures are appropriate but for an easier interpretation authors could make the following changes:

Table 1: please eliminate the double dot at the end of the table caption

Table 3: add a space after the table (before line 212); Please modify the table caption “examples of alternative definitions for biosecurity” instead of “example of biosecurity terms” and use the correct form “Definition for biosecurity” as title of the second cell of the table.

Table 2 N.11: Authors should replace “vermin control” with “pest control” (in the table and throughout the text)

Table 5: Please reformat the table (be careful to spaces, use of bolding) and correct the names of biosecurity sections N.1 and N.12 as written in Table 2.

The cited references are mostly recent and relevant but some of them should be edited e.g.

lines 732-733: doi does not work and authors could cite this Regulation (EU) in this way:

Regulation (EU) No 2016/429 of the European Parliament and of the Council of 9 March 2016 on Transmissible Animal Diseases and Amending and Repealing Certain Acts in the Area of Animal Health (‘Animal Health Law’) O J L 84, 31.3.2016. Available online: https://eur-lex.europa.eu/legal-content/EN/TXT/?uri=uriserv%3AOJ.L_.2016.084.01.0001.01.ENG

Lines 852-853: authors could cite this Regulation (EU) in this way:

Regulation (EU) No 2016/429 of the European Parliament and of the Council of 9 March 2016 on Transmissible Animal Diseases and Amending and Repealing Certain Acts in the Area of Animal Health (‘Animal Health Law’) O J L 84, 31.3.2016. Available online: https://eur-lex.europa.eu/legal-content/EN/TXT/?uri=uriserv%3AOJ.L_.2016.084.01.0001.01.ENG

After changing the reference, please change also the quote in Table 3 N.1.

The conclusions are consistent with the arguments presented and offer ideas for future studies e.g. epidemiological ones.
Taking all this into account the work fits the journal aims. 

Reviewer 2 Report

The technical note by Pedersen et al. is a very nice work which transparently explains the development of the BAF-SD framework. The choices they make are well described and justified. The transparant approach, coupled by making the framework public is admirable, comparing to other systems which tend to be a "black box". Therefore the framework provides a structural approach to comprehensevely assess the biosecurity level of dairy farms with regard to the introduction and establishment (! well justified) of SD. It is hoped this will be of interest in stagnating SD control in Danmark & other countries.

The description of the literature provides a state of the art interpretation of current knowledge on Salmonella infection in cattle.

The authors have choisen to highlight the (increasing) importance of herd size and complexity in farm structure in the "simple summary" and abstract of this technical note. Although I am of the same opinion of the authors that nowadays, these elements (herd size,...) demand higher levels of biosecurity, this claim is only lightly mentioned in the introduction (line 46-47) and party explained in the literature review part (larger herds, more movements). This is a non critical remark (I was triggered by the start of the abstract). This might be solved to refer to some epidemiological studies related to herd size, if present (perhaps also in section 3.3.3 'establishment', where other factors like concurrent infectious are described.

The methodology is clear using the six phases and includes an evaluation of assessors, and the authors include guides in the excel to conduct the assessment.

It is remarkable the within the 12 biosecurity sections, "introduction of cattle" is not a seperate item, but incorporated in items no 2 and 3. Perhaps this choice is related to the systematic control measures are in place in the Danish system?

I belief tables should be interpreted on their own, therefore the term MeSH can be written full in the description of the title of table 1?

As because of a 'technical note', the discussion of the submitted papers questions its methodology, the justification of the framework in terms of controlling SD is less discussed, and provides an opportunity for further work...

Finally, the authors made a very fine work, which I belief is valuable to be published. My sincere apologies to the authors and editor for my delay in response.

Reviewer 3 Report

Interesting proof of concept study.  Might have been better to delay publication until you have externally validated assessments conducted but useful methodological study. Could describe typical Danish dairy farm structure/management (or those used) for external readers as background.

Suggest have a native English speaker read the manuscript - a few irritating sentence structures and inappropriate use of words, e.g. why. Style is very verbose, could do with substantial abbreviation.
